# Auxin Transporter OsPIN1b, a Novel Regulator of Leaf Inclination in Rice (*Oryza sativa* L.)

**DOI:** 10.3390/plants12020409

**Published:** 2023-01-15

**Authors:** Yanjun Zhang, Shaqila Han, Yuqing Lin, Jiyue Qiao, Naren Han, Yanyan Li, Yaning Feng, Dongming Li, Yanhua Qi

**Affiliations:** 1Key Laboratory of Herbage & Endemic Crop Biology of Ministry of Education, Inner Mongolia Key Laboratory of Herbage & Endemic Crop Biotechnology, School of Life Sciences, Inner Mongolia University, Hohhot 010030, China; 2State Key Laboratory of Plant Physiology and Biochemistry, College of Life Sciences, Zhejiang University, Hangzhou 310058, China; 3State Key Laboratory of Rice Biology, China National Rice Research Institute, Chinese Academy of Agricultural Sciences, Hangzhou 310006, China; 4College of Life Science and Technology, Inner Mongolia Normal University, Hohhot 010022, China

**Keywords:** auxin, auxin transporter, BR, leaf inclination, *OsPIN1b*, rice

## Abstract

Leaf inclination is one of the most important components of the ideal architecture, which effects yield gain. Leaf inclination was shown that is mainly regulated by brassinosteroid (BR) and auxin signaling. Here, we reveal a novel regulator of leaf inclination, auxin transporter OsPIN1b. Two CRISPR-Cas9 homozygous mutants, *ospin1b-1* and *ospin1b-2,* with smaller leaf inclination compared to the wild-type, *Nipponbare* (WT/NIP), while overexpression lines, *OE-OsPIN1b-1* and *OE-OsPIN1b-2* have opposite phenotype. Further cell biological observation showed that in the adaxial region, *OE-OsPIN1b-1* has significant bulge compared to WT/NIP and *ospin1b-1*, indicating that the increase in the adaxial cell division results in the enlarging of the leaf inclination in *OE-OsPIN1b-1*. The OsPIN1b was localized on the plasma membrane, and the free IAA contents in the lamina joint of *ospin1b* mutants were significantly increased while they were decreased in *OE-OsPIN1b* lines, suggesting that OsPIN1b might action an auxin transporter such as AtPIN1 to alter IAA content and leaf inclination. Furthermore, the *OsPIN1b* expression was induced by exogenous epibrassinolide (24-eBL) and IAA, and *ospin1b* mutants are insensitive to BR or IAA treatment, indicating that the effecting leaf inclination is regulated by *OsPIN1b*. This study contributes a new gene resource for molecular design breeding of rice architecture.

## 1. Introduction

Lamina joint inclination (leaf inclination/angle) between the lamina and vertical culm is an important agronomic trait in rice (*Oryza sativa* L.), which is considered to reflect the status of rice cultivation and grain yield [1]. In rice, leaf erectness has an important effect on the capture of sunlight and CO_2_ diffusion efficiency. The reduction in leaf inclination enables a single leaf to capture sunlight on both sides, and reduces mutual shielding of leaves, improves light transmittance, and has stronger solar absorption efficiency, which is more suitable for high-density planting [2]. The lamina joint is a ring-shaped tissue outside the joint of the leaf blade and leaf sheath, which influences orientation of extension and leaf inclination, and provides mechanical strength for the shape of leaf inclination. The cells on the adaxial side of the lamina joint play an essential role in adjusting leaf inclination [3,4]. For instance, the increased growth of cells on the adaxial side of the lamina joint in *ili1-D* causes an enlarged lamina inclination phenotype [5]. Conversely, CYC U4;1 promotes leaf erectness by controlling the abaxial sclerenchyma cell proliferation [6].

By previous report, BR biosynthesis or signaling transduction contributes significantly to regulating the leaf inclination. Deficient mutant of *OsDWARF4*, a rate-limiting gene in BR biosynthesis, displayed erect leaf inclination phenotype and enhanced grain yields [2]. The increased leaf erectness phenotypes also were found in several loss of function mutants of BR biosynthesis-related genes, including *brassinosteroid-deficient dwarf1* (*brd1*, deficiency of *OsDWARF*), *ebisu dwarf* (*d2*, deficiency of *D2/CYD90D2*), and *dwarf11* (*d11*, deficiency of *D11/CYP724B1*) [7,8]. Furthermore, BR signaling defective mutant, *d61*, was also similar to mutant of rice *BRASSINOSTEROID INSENSITIVE 1* (*OsBRI1* encodes a putative BR receptor kinase), which have erect leaves [4]. The lamina joint bending of RNAi mutant of the transcription factor OsBZR1 involved in the BR signaling pathway, was reduced [9].

In addition to brassinosteroid (BR), auxin signaling also plays a crucial role in controlling leaf inclination, which may be closely related to differential cell elongation in adaxial and abaxial in the lamina joint [6,10,11,12]. In auxin signaling, auxin early response genes, including the *OsIAA*, *OsGH3,* and *OsSAUR* family were reported to be related to regulating leaf inclination. *OsIAA1*-overexpressed plant increased the lamina joint bending and reduced the plant height [11,13,14,15,16]. Further study demonstrated that an EMBRYONIC FLOWER 1-like (EMF1-like) protein encoded by *dwarf and small grain size 1* (*DS1*), physically interacted with OsARF11 to positively co-regulate plant height and leaf angle [17]. Recent study indicated that the auxin-induced expression of *ILA1* gene is dependent on OsARF6 and OsARF17 binding to the promoter region of *ILA1* [18]. Loss-of-function double *osarf6 osarf17* mutants displayed reduced secondary cell wall thickness of lamina joint sclerenchyma cells due to the decreased expression level of *ILA1*, affecting the support weight of the flag leaf blade, finally an enlarged flag leaf angle and reduced grain yield under dense planting conditions. As the major biosynthetic pathway of indole-3-acetic acid (IAA), the indole-3-pyruvate (IPA) pathway, is catalyzed by the TRYPTOPHAN AMINOTRANSFERASE of ARA-BIDOPSIS (TAA) and YUCCA (YUC) families. *FISH BONE* (*FIB*) encodes a homologue of the TAA protein, which plays a negative role in leaf inclination [19]. Loss-function of *FIB* leads to decreased IAA levels and altered auxin polar transport activity, resulting in increased leaf inclination and smaller leaves.

In addition, polar auxin transport (PAT) regulates auxin distribution, which is largely dependent on the three protein families, including influx carrier AUXIN/LIKE AUXIN (AUX/LAX), efflux carrier PIN-FORMED (PIN), and bidirectional carrier ATP-binding cassette family B (ABCB)/P-glycoprotein (PGP) [20,21,22,23,24,25]. The PIN proteins play crucial roles in direction and rate of auxin flow [26]. However, the biological function of auxin transporters regulating leaf inclination is not yet reported. In *Arabidopsis thaliana*, PIN1 is the first reported auxin efflux carrier, and the *atpin1* pin-formed inflorescence and defective development of vascular tissue [20,27,28,29,30]. Among the twelve members of *PIN* in rice, *OsPIN1* is composed of four members, *OsPIN1a*, *OsPIN1b*, *OsPIN1c*, and *OsPIN1d* [31]. It was reported that *OsPIN1b* functions in regulating tillering, adventitious root emergence, and seminal roots elongation [32,33]. *OsPIN2*-mediated auxin transport from the shoot to the root-shoot junction, resulted in the reduced plant height, the increased tiller numbers, and the increased tiller angle [34]. OsPIN5b is located in endoplasmic reticulum (ER) and involves in intracellular auxin transport to regulate tiller number, panicles length, and yield [35,36]. *OsPIN9* was involved in the development of tiller bud [37]. *OsPIN3t* (*OsPIN10a*/*OsPIN3a*) was connected with auxin polar transport and drought response [31,38,39]. In this study, we revealed that the homozygous *ospin1b* mutants have smaller lamina inclination, whereas *OsPIN1b* overexpression lines have larger lamina inclination compared to WT/NIP. Additionally, *OsPIN1b* responded to BR and auxin signaling, suggesting that *OsPIN1b* plays a role in regulating lamina inclination under crosstalk between both signaling.

## 2. Results

### 2.1. Identification of ospin1b Mutants and OsPIN1b Overexpression Lines

By previous reports, *OsPIN1b* showed that it regulates root, shoot inflorescence development, and tillering [32,33]. However, the other roles of *OsPIN1b* in growth and development of rice remain unknown. In order to investigate thoroughly the biological function of *OsPIN1b*, we constructed two independent mutant lines of *ospin1b*, *ospin1b-1,* and *ospin1b-2* by the CRISPR-Cas9 genome editing system. Two specific guide RNA (gRNA) sequences were designed in the first exon, acting as gene edit targets (Appendix A). The gRNA1 targeting sequence deleted one cytosine “C” at 418 bp of *OsPIN1b* open reading frame in *ospin1b-1*, while it inserted one cytosine “C” at 491 bp in the targeting sequence of gRNA2, and resulted in the variation in amino acid sequence and premature termination of translation. For *ospin1b-2*, in gRNA1 targeting sites inserted one adenine “A” at 418 bp, and inserted one thymine “T” at 494 bp in the targeting sequence of gRNA2, which leaded to similar consequences with *ospin1b-1* (Figure 1A, Appendix A). Furthermore, *OsPIN1b* overexpression lines were constructed using *pCAMBIA1300-sGFP* (Figure 1B), and two homozygous *OE-OsPIN1b-1* and *OE-OsPIN1b-2,* which have a 1.5-fold and 2.5-fold expression level compared to *OsPIN1b* of WT/NIP by RT-qPCR analysis (Figure 1E), were used in subsequent studies. By statistical analysis, the leaf inclinations in *ospin1b* mutants were reduced ~30%, while in *OsPIN1b,* overexpression lines were increased ~140% compared to WT/NIP (Figure 1C,D). These results suggest that *OsPIN1b* might contribute to regulating leaf inclination.

### 2.2. Characterization of Complementary ospin1b Lines

To further confirm the relationship between *OsPIN1b* and leaf inclination phenotype in rice, the complementary *ospin1b* lines, *C-ospin1b-1* and *C-ospin1b-2,* were constructed by transforming *35S:OsPIN1b-sGFP* into *ospin1b-1* or *ospin1b-2*. Compared with the *ospin1b* mutants, the flag leaf inclination of *C-ospin1b-1* or *C-ospin1b-2* were significantly increased (Figure 2A,B), indicating that *OsPIN1b* rescued the phenotype of less leaf inclination of *ospin1b-1* or *ospin1b-2*. The results further provide genetic evidence for *OsPIN1b* controlling leaf inclination.

### 2.3. OsPIN1b Constitutively Expressed in Each Tissue Including Lamina Joint

To further investigate whether *OsPIN1b* is involved in lamina joint development to regulate leaf inclination, the expression pattern of *OsPIN1b* was observed using *ProOsPIN1b-GUS* transgenic rice lines. GUS staining results show that *OsPIN1b* is expressed in each tissue, including roots, stem, leaf, leaf sheath, flower, and seed (Figure 3A–G); qRT-PCR results also showed that *OsPIN1b* was widely expressed in each organ or tissue, which was consistent with GUS staining. Additionally, the expression level of *OsPIN1b* was significantly increased in leaf and young panicle (Figure 3H). Furthermore, *OsPIN1b* was expressed in lamina joint from one to four weeks (Figure 3I), suggesting that *OsPIN1b* might be involved in the regulation of each stage of leaf inclination in rice.

### 2.4. OsPIN1b Localized on Plasma Membrane and OsPIN1b Reduced the Free IAA Accumulation

In Arabidopsis, hydrophilic analysis of PIN demonstrates that there is a fairly high degree of similarity in both transmembrane hydrophobic domains located at the N- and C-terminus of the proteins, while high differentiation in the central hydrophilic region [40]. Additionally, previous research established that OsPIN proteins have a similar structure to AtPIN [31]. To investigate the function of OsPIN1b, we performed transmembrane structural domain prediction of OsPIN1b. The results show that OsPIN1b possesses a central segment with a long hydrophilic loop and two transmembrane hydrophobic regions in the N-terminal and C-terminus of the protein, which are consistent with the structure of a PIN carrier protein (Appendix A). To further confirm whether OsPIN1b is located on the plasma membrane, *35S: OsPIN1b-sGFP* and the plasma membrane marker pm-rb*CD3-1008* were transiently co-expressed and transformed into the leaf epidermal cells of *Nicotiana benthamiana* and rice protoplasts [41]. Both experiments indicate that OsPIN1b is actually localized on the plasma membrane (Figure 4A,B). Taken together, these results suggest that OsPIN1b might have an auxin transport function similar to AtPIN1, which was characterized as a first putative auxin export carrier [29].

In order to explore if *OsPIN1b* affects the leaf inclination through altering of *OsPIN1b*-mediated IAA contents, we analyzed free IAA contents in the lamina joints of WT/NIP, *ospin1b* mutants, and *OsPIN1b* overexpressed lines at one week of the seedling stage. The IAA content of lamina joint of *ospin1b-1* and *ospin1b-2* was increased by 32 and 220%, respectively, compared with the WT/NIP, while the IAA content in lamina joint of *OE-OsPIN1b-1* and *OE-OsPIN1b-2* were decreased compared with WT/NIP (Figure 4C). The results suggest that the *OsPIN1b* functions in decreasing the free IAA accumulation in lamina joint, which is similar with *OsARF19* [42].

### 2.5. OsPIN1b Enhances Adaxial Cell Division of Pulvinus to Increase Leaf Inclination

Increasing evidences show that the altered pulvinus development effects leaf inclination [4,43]. In order to reveal how *OsPIN1b* controls leaf inclination on the cell level, the flag leaf inclination and the adaxial and abaxial surface of pulvinus were observed and measured in WT/NIP, *ospin1b-1*, *ospin1b-2*, *OE-OsPIN1b-1,* and *OE-OsPIN1b-2*. The results show that the flag leaf inclination of the mutant *ospin1b-1* and *ospin1b-2* was smaller, and the length of the adaxial side of pulvinus in *ospin1b-1 or ospin1b-2* was shorter, whereas flag leaf inclination and adaxial length of pulvinus showed opposite phenotypes in *OE-OsPIN1b*, compared to in WT/NIP (Figure 5A–C). To further confirm the pulvinus morphology, the adaxial side of pulvinus in WT/NIP, *ospin1b-1* and *OE-OsPIN1b-1* were observed using scanning electron microscopy. As showed in Figure 5D, the adaxial region of *OE-OsPIN1b-1* has significant bulge compared to WT/NIP and *ospin1b*-1. These results further support that the enlarging leaf inclination in *OE-OsPIN1b-1* is due to the increase in the adaxial cell division of its pulvinus. By previous reports, the asymmetric proliferation and expansion of adaxial and abaxial cell of pulvinus leaded the variation in leaf inclination, which were induced by BR or IAA frequently, suggesting that *OsPIN1b*-mediated leaf inclination variation might also be associated with BR or IAA [5,6,42,44].

### 2.6. BR or Auxin Induces OsPIN1b Expression, and the Expression of the Genes Related Both Signaling Reduces in ospin1b Mutants

BR and auxin are the two important phytohormones affecting leaf inclination by previous report. Most of the mutants related auxin signaling with altered leaf inclination also showed BR response. To clarify whether *OsPIN1b* regulating leaf inclination is also involved in BR and auxin signaling, first, the expression level of *OsPIN1b* in WT/NIP seedlings were tested with the different time under these phytohormones treatments. The results show that *OsPIN1b* was induced by BR and IAA, indicating that the expression of *OsPIN1b* in rice might be regulated by these signaling pathways (Figure 6A). Then, the expression levels of the five genes related BR and auxin signaling or biosynthesis, *OsBRI1*, *D2*, *D11*, *OsARF19*, and *OsIAA1* were detected in WT/NIP, *ospin1b*, and *OE-OsPIN1b*. *OsBRI1*, a membrane-bound receptor kinase, with leucine rich repeat (LRR) perceives BR signaling [45,46,47,48]. Compared to wild-type, loss-of-function mutants of *OsBRI1*, *d61-1* and *d61-2* are insensitive to BR, with erect leaves and dwarf culms [4,49]. Shrinking leaf inclination phenotypes were observed in deletion mutants of BR biosynthesis genes *D2 (CYP90D2)* and *D11 (CYP724B1)* [8,50]. Compared with the wild type, the *OsBRI1*, *D2* and *D11* expression levels in the lamina joint of *ospin1b* mutants was significantly down-regulated, and the opposite trend was observed in *OsPIN1b* overexpression lines (Figure 6B), suggesting that *OsPIN1b* regulating leaf inclination was involved in BR signaling. In addition, the expressions of the auxin early response gene *OsIAA1* and *OsARF19* encoding auxin response factor, which was included in auxin signaling (Figure 6B), was significantly up-regulated in *OsPIN1b* overexpression lines; and their overexpression lines with increased leaf inclination [14,42] were consistent of *OE*-*OsPIN1b* lines, implying that *OsPIN1b* might also participate in the auxin signal transduction pathway.

### 2.7. The Decreased Sensitivity to BR and IAA in ospin1b Mutants

To further understand if *OsPIN1b* controls leaf inclination through the BR and auxin pathway, the lamina inclinations of the WT/NIP, *ospin1b* mutants and *OE-OsPIN1b* lines were measured under BL, IAA, and BL + IAA treatments, respectively. Phenotypic observation showed that under BL treatment, leaf inclination of WT/NIP, *ospin1b* mutants and *OE-OsPIN1b* lines were more enlarged than control, with the increasing in BL concentration (Figure 7A). In particular, leaf inclination of WT/NIP increased by 264%, *ospin1b* mutants increased by 173%, and *OE-OsPIN1b* increased by 460% under 1 μM of BL concentration compared with non-BL treatment (Figure 7B). These results suggest that the sensitivity of *ospin1b* mutants to BR was reduced in terms of leaf inclination, which was caused by the loss of *OsPIN1b* function. On the other hand, the leaf inclination of WT/NIP, *ospin1b* mutants, and *OE-OsPIN1b* lines were effectively increased with the increase in IAA concentration (Figure 7C). Under 100 μM IAA, the mean leaf inclination of WT/NIP, *ospin1b* mutants, and *OE-OsPIN1b* lines were increased by nearly 102%, 88%, and 129%, respectively, compared with each untreated sample (Figure 7D). The leaf inclination of *ospin1b* mutants also showed that the decrease in sensitivity to IAA. Furthermore, the leaf inclination of each line to the application of both BL and IAA was much larger than single BL or IAA (Figure 7E,F). Distinctly, after applied 1 + 100 μM of BL and IAA together, the leaf inclinations were increased nearly 629% and 738% in WT/NIP and *OE-OsPIN1b*, while the increase is less than 540% in *ospin1b* mutants. Overall, these results suggest that *OsPIN1b* might participate in crosstalk between BR and auxin signaling in regulating leaf inclination.

In addition, according to a previous BR sensitivity report, coleoptile elongation and root length were promoted by BR treatment in rice [4]. Hence, we compared the length of coleoptiles and roots in WT/NIP, *ospin1b,* and *OE-OsPIN1b* lines under BR treatment (Appendix A). All lines increased the coleoptile length and shortened the root length, and showed a dose-dependent pattern under 24-eBL treatment. However, the *ospin1b* mutants was less sensitive to 24-eBL than WT/NIP or the *OsPIN1b* overexpression lines, especially in their roots in response to BR. Taken together, the above data further confirm that the other biofunctions of *OsPIN1b* are also related to BR signaling.

## 3. Discussion

### 3.1. OsPIN1b Plays a Positive Role in Enlarging Leaf Inclination via Regulating Auxin Transport in Rice

The distribution and level of auxin in plant tissues play an important role in plant growth and development [51]. As an auxin transporter, the *PIN* family mainly facilitates auxin distribution and flow direction [26,27,28,52,53]. The loss of function of auxin transporters may lead to abnormal plant phenotypes [54]. This study revealed that the IAA content in lamina joint of *ospin1b-1* and *ospin1b-2* was significantly increased, while decreased in *OE-OsPIN1b* lines (Figure 4C). The leaf inclination in *ospin1b* mutants was reduced, while in *OsPIN1b*, overexpression lines were increased compared to WT/NIP (Figure 1C,D). The above results imply that *OsPIN1b* positively regulates leaf inclination through altering auxin transport. Previous studies of auxin transporters in rice mostly focused on root development and tiller, our study demonstrated that OsPIN1b also plays a role in regulating rice leaf inclination [24,25,33]. Recently, the *Gmpin1abc* and *Gmpin1bc* mutant edited by CRISPR-Cas9 showed a compact architecture with smaller petiole angles than wild-type plants, which is similar with our results [55].

### 3.2. OsPIN1b Participates in Adaxial Cell Enlargement of Lamina Joint through Complex Regulatory Mechanism of Auxin and BR

In our study, the enlarging of *OE-OsPIN1b* leaf inclination is caused by the increasing in adaxial cell numbers in the lamina joint, while the phenotype in the *ospin1b* mutant line was opposite (Figure 5C). Previous studies showed that BR and IAA both regulate leaf inclination by changing the unbalanced development between the adaxial and abaxial cells of the lamina joint. In rice, *INCLINATION1 (ILI1)* and *ILI1 binding bHLH (IBH1)* antagonistically regulate the adaxial cell elongation in lamina joint by interacting with OsBZR1, a transcription factor involved in BR signaling [5]. The U-type cyclin *CYC U4*;*1* was highly expressed in the lamina joint. The proliferation of sclerenchyma cells at the abaxial side of lamina joint is regulated by the BR regulation pathway, which leads to the erect leaves [6]. *Leaf inclination1* (*LC1)* encodes an indole-3-acetate (IAA) amide synthetase OsGH3-1, whose functional gain mutant *lc1-D* exaggerated leaf angles due to increased cell elongation on the paraxial surface of the lamina joint [16]. Over-expression lines of *auxin response factor 19* (*OsARF19*) in rice, shows an enlarged lamina inclination due to increased adaxial cell division [42]. These reports are similar to the *OsPIN1b* results, suggesting that they have a certain correlation in regulating lamina inclination through altering adaxial cell division.

*OsPIN1b* can be significantly induced by BR and IAA (Figure 6A). It is proven that phytohormone BR and IAA are the key reasons for *OsPIN1b* affecting leaf inclination. Furthermore, we analyzed the expression of BR or IAA biosynthesis and signal transduction-related genes in each transgenic line of *OsPIN1b*, and found that the expression of these genes was significantly inhibited in the mutants, but induced in the *OsPIN1b* overexpressing line, which was consistent with the insensitive response of *ospin1b* to BR and auxin (Figure 6B and Figure 7A–F). This is similar to the previous research that the BR signal transduction receptor BRI1 positively regulates the leaf inclination of rice, and the leaf inclination increases in the overexpression line of auxin response factor OsARF19 [4,42]. In addition, previous studies found that auxin stimulates the response of BR by increasing the levels of the brassinosteroid receptor BRI1 [11]. OsARF19 increases leaf inclination by positive regulation of *OsGH3-5* and *OsBRI1* [42]. Aiming at regulation of leaf inclination, OsARF4 (auxin response factor 4) plays a role between auxin and BR signaling pathways [56]. In this study, 24-eBL greatly promoted the leaf inclination, while IAA slightly enlarged the leaf inclination with the increase in their concentration (Figure 7A–F). BR induced much higher leaf inclination than IAA, and a synergistic effect was observed between BR and IAA, which accords with our results [12,43]. These results prove that crosstalk occurs between auxin and BR during *OsPIN1b,* regulating lamina joint development and leaf inclination.

### 3.3. ospin1b Reduced the Plant Height and Leaf Inclination and Formed the Ideal Plant Architecture

Rice is one of the most important food crops, feeding more than half of the world’s population. With the steady growth of population, human demand for rice will increase rapidly, which requires the cultivation of excellent rice varieties with ideal plant configuration. Just as the first “green revolution”, the grain yield was significantly improved by planting lodging-resistant wheat and rice semi-dwarf varieties [57]. Thus, crop plants with desirable architecture, such as lodging-resistant semi-dwarf varieties are able to produce much higher grain yields. Leaf angle is one of the most important plant architecture parameters affecting light interception, photosynthetic efficiency, and planting density [1,2]. Our research found that *ospin1b* lines reduced leaf inclination (Figure 1C,D and Figure 5A,B) and decreased plant height (Appendix A), which is similar to the optimized architecture of rice. Thus, our study provides a novel perspective on the mechanism that auxin regulates the leaf inclination.

## 4. Materials and Methods

### 4.1. Plant Materials and Growth Conditions

Rice seeds of wild-type, mutant, and transgenic lines were submerged (soaked) in water for 3 days in darkness of 37 °C, then the most uniformly germinated seeds were scattered on a floating net. Rice seedlings were cultured in nutrient solution (pH 5.5) [58] and grown under a 12/12 h light/dark cycle in a greenhouse of chamber at 28 °C/22 °C for 30 days.

For exogenous phytohormone treatments of *OsPIN1b* expression, 7-day-old WT/NIP rice seedlings were treated with different exogenous phytohormone (10 μM of IAA, 10 μM of 24-eBL, epibrassinolide). For the sensitivity test of BR, the WT/NIP, *ospin1b* mutants, and *OsPIN1b*-overexpression lines were cultured in nutrient solution (pH 5.5) supplemented with 0.01, 0.1, 1 µM 24-eBL for 7 days in a greenhouse at 26 °C for 24 h dark per day, respectively. The length of root and coleoptile per line was measured and photographed.

### 4.2. Construction and Identification of ospin1b Mutants

CRISPR-Cas9-mediated genome editing technology was used to obtain *ospin1b* mutant lines. Cas9/gRNA target site selection and vector construction were conducted according to previous reports [59]. The *OsPIN1b-pRGEB32* vector was introduced into Agrobacterium strain EHA105 and transformed into WT/NIP calli, as described previously [60]. Homozygous *ospin1b* mutants were screened by PCR analysis of the Cas9 label and DNA sequencing of the *OsPIN1b* specific editing site. The primer sequences used for plasmid construction and mutant identification are listed in Appendix A.

### 4.3. Construction and Transformation of Binary Vectors

The open reading frame of *OsPIN1b* was amplified from the cDNA of WT/NIP, and the 2.6 kb promoter of the *OsPIN1b* was amplified from WT/NIP genomic DNA. *35S:OsPIN1b-sGFP* and *ProOsPIN1b:GUS* vectors were constructed as described previously [25]. These vectors were introduced into *Agrobacterium* strain EHA105 and transformed into WT/NIP calli. Primer sequences are listed in Appendix A.

### 4.4. Subcellular Localization of OsPIN1b

The full-length coding region of *OsPIN1b* was inserted into a binary *pCAMBIA1300* vector, which was labeled with synthetic green fluorescent protein (sGFP). The *35S:OsPIN1b-sGFP* and plasma membrane marker *pm-rbCD3-1008* were co-transfected into the *Nicotiana bentamiana* epidermal cells by Agrobacterium transformation and the isolated rice protoplasts by polyethylene glycol/calcium transfection and incubated overnight [61]. The fluorescent signals of the expressed proteins were observed by two-photon fluorescence microscopy (Zeiss LSM710; Carl Zeiss, Oberkochen, Germany), as described previously [24].

### 4.5. β- Glucuronidase (GUS) Staining

Each tissue (seeds germinated, 3 d; roots, 7 d; stems and leaves, 14 d; flowers, and 2 month of *ProOsPIN1b:GUS* transgenic rice were incubated in GUS staining buffer for 2 h at 37 °C, and then these tissues were soaked in 95% ethanol to remove chlorophyll and surface dye. Images of GUS-stained tissues were observed by stereomicroscope (Leica MZ95 microscope, Leica, Wetzlar, Germany).

### 4.6. RNA Extraction, RT-PCR, and Real-Time Quantitative RT-PCR (qRT-PCR) Analyses

Total RNA was extracted from various tissues of rice WT/NIP seedlings, lamina joint of different periods, and different exogenous phytohormone-treated plants; seedlings of wild-type, *ospin1b* mutant, and *OE-OsPIN1b* by using the TIANGEN RNAprep pure Plant Kit (TIANGEN BIOTECH, Beijing, China) according to the manufacturer’s protocol. RNA was quantified using a spectrophotometer (NaNoDrop 1000; Thermo Scientific, United States). Total RNA (2 μg) was used to synthesize cDNA by using Hifair^®^ Ⅱ1 st Strand cDNA Synthesis SuperMix for qPCR (gDNA digester plus), and qRT-PCR was conducted using Hieff^®^ qPCR SYBR Green Master Mix (No Rox) (Yeasen Biotechnology, Shanghai, China) in a LightCycler^®^ 480 (Roche, Basel, Switzerland). The cycle conditions were as follows: 95 °C for 5 min, one cycle; 95 °C for 10 s, 58 °C for 20 s, and 72 °C for 20 s, 40 cycles; qRT-PCR was performed using three independent experiments with biological triplicates for each sample. *OsACTIN* (Os03g50885) was used as an internal control to calculate the fold change in expression. The primer sequences used for qRT-PCR are listed in Appendix A.

### 4.7. Exogenous Hormone Response Assay

Epibrassinolide (24-eBL, designated as BL; Solarbio Science and Technology, Beijing, China) was dissolved in DMSO, and indoleacetic acid (IAA; Sigma, Darmstadt, Germany) was in ethanol, to proper concentrations as storage solutions. For the hormone treatments of leaf inclination, WT/NIP, *ospin1b* mutants, and *OsPIN1b*-overexpression lines were cultured in darkness for 7 days in a exogenous hormone-free nutrient solution, and then the lamina joints were cut with a length of about 3 cm and soaked in ddH_2_O containing 0, 0.01, 0.1, and 1 µM 24-eBL (0, 1, 10, and 100 µM IAA; 0 + 0, 0.01 + 1, 0.1 + 10, and 1 + 100 µM 24-eBL + IAA) for 3 days under same conditions, respectively. The leaf inclination in each group was measured and photographed. All experiments were independently repeated three times.

### 4.8. Scanning Electron Microscopy (SEM)

Scanning electron microscopy was performed as described previously [44]. About 1 cm lamina joints of flag leaf of a two-month plant were excised from WT/NIP, *ospin1b* mutants and overexpression lines of rice. Samples were observed with an S-3000N scanning electron microscope (Hitachi, Tokyo, Japan).

### 4.9. Endogenous IAA Contents Analysis by HPLC

The lamina joints (about 1 cm in length, 100 mg) of 7-day-old seedlings were ground to powder in liquid nitrogen. Extracting IAA and measuring its content refer to the previous research [62] using a Rigol L3000 high-performance liquid chromatograph with a C18 reversed-phase chromatographic column (250 mm × 4.6 mm, 5 μm). These Samples were analyzed by HPLC-electrospray ionization-tandem mass spectrometry at Suzhou Keming Biotechnology Company (Suzhou, China).

### 4.10. Accession Numbers

Sequence data from this article can be found in the NCBI Database or Rice Genome Annotation Project under the following accession numbers: *OsPIN1b*, LOC_Os02g50960; *OsBRI1*, *LOC_Os01g52050*; *D2*, *LOC_Os01g10040*; *D11*, *LOC_Os04g39430*; *OsARF19*, *LOC_Os06g48950*; *OsIAA1*, *LOC_Os01g08320*; and *OsACTIN*, *LOC_Os03g50885.*

### 4.11. Primer Sequences

The primers used are shown in Appendix A.

## Figures and Tables

**Figure 1 plants-12-00409-f001:**
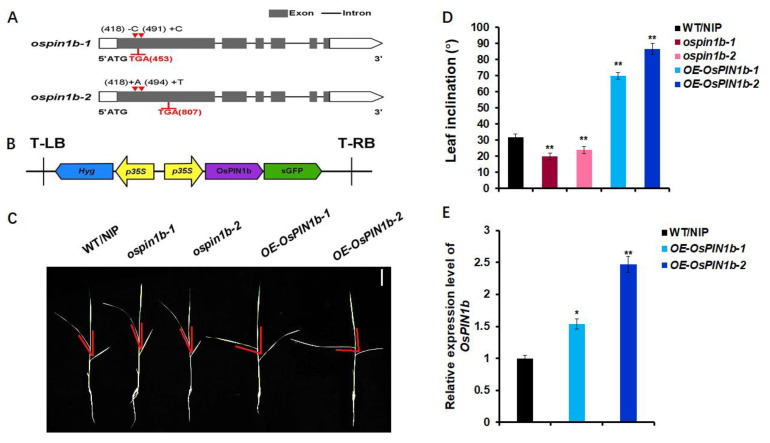
Characterization of *ospin1b* mutants and *OsPIN1b* overexpression lines. (**A**) Construction of two *ospin1b* mutants by CRISPR-Cas9. Gray boxes, black lines, and white boxes represent exons, introns, and untranslated region, respectively; + and ▼ indicates the insertion sites in *ospin1b* mutants, − and ▼ indicates the deletion site in *ospin1b* mutants. ATG and TGA presents start codon and stop codon. (**B**) Structure of *OsPIN1b* overexpression vector *pCAMBIA1300-sGFP*. (**C**,**D**) Phenotypes and statistics analysis of the leaf inclination in the second leaf of WT/NIP, *ospin1b* mutants, and *OsPIN1b* overexpression lines for 3-week-old seedling. The angle between the red lines represents the leaf inclination. The statistical data are mean ± SD (n = 3) and * indicates the significant difference among WT/NIP, *ospin1b*, and *OE-OsPIN1b* (** *p* < 0.01; Student’s *t*-test). Scale bar = 5 cm. (**E**) qRT-PCR analysis of *OsPIN1b* expression in WT/NIP and *OsPIN1b* overexpression lines. Three independent biological replicas were performed here. *OsACTIN* was used as an internal control. The data are mean ± SD (n = 3) and * indicates the significant differences between WT/NIP and *OE-OsPIN1b* (* *p* < 0.05, ** *p* < 0.01; Student’s *t*-test).

**Figure 2 plants-12-00409-f002:**
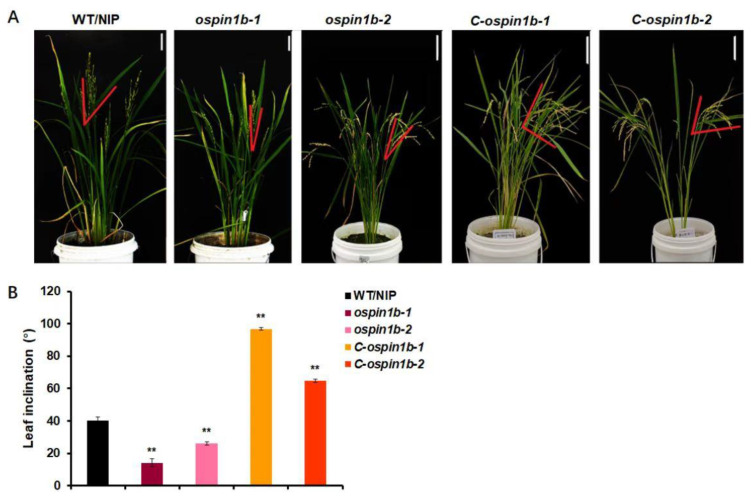
Identification of complementary *ospin1b* lines. (**A**,**B**) The phenotype and statistics analysis of flag leaf inclination in WT/NIP, *ospin1b-1*, *ospin1b-2*, complementary *ospin1b* lines for 4-month-old, *C-ospin1b-1*, and *C-ospin1b-2*. The angle between the red lines represents the flag leaf inclination. The statistical data are mean ± SD (n = 3) and ** indicates significant differences between WT/NIP, *ospin1b-1*, *ospin1b-2*, *C-ospin1b-1,* and *C-ospin1b-2* (** *p* < 0.01; Student’s *t*-test). Scale bar = 10 cm.

**Figure 3 plants-12-00409-f003:**
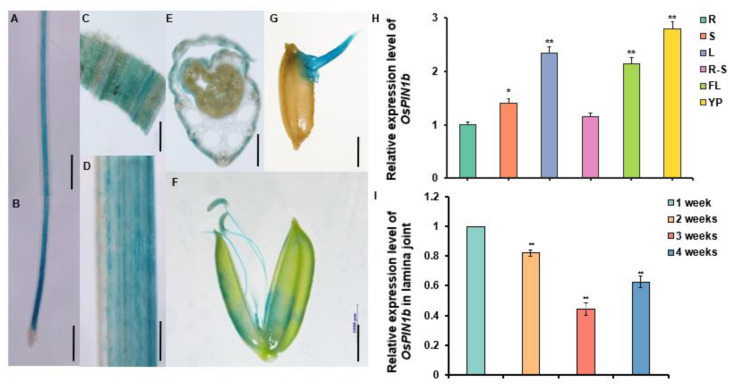
Expression pattern of *OsPIN1b* gene. GUS staining in each tissue of *ProOsPIN1b: GUS* transgenic rice. Ten biological replicas were analyzed for each tissue. Scale bars = 1 mm. (**A**) Maturation region of primary root, (**B**) elongation zone, meristem zone, and root cap of primary root, (**C**) stem, (**D**) leaf, (**E**) transverse section of leaf sheath, (**F**) floral organ, (**G**) the germinated seed for 3 days. (**H**) qRT-PCR analysis of relative expression level of *OsPIN1b* in different tissues of WT/NIP. R: root; S: stem; L: leaf; R-S: root-stem junction; FL: flag leaf; YP: Young panicle. *OsACTIN* was used as an internal control. The data are mean ± SD (n = 3) and asterisks indicate significant differences in different tissues (* *p* < 0.05, ** *p* < 0.01; Student’s *t*-test). (**I**) qRT-PCR analysis of relative expression level of *OsPIN1b* in lamina joint at different stages of WT/NIP from one to four weeks. The three independent biological repeats were performed in the qRT-PCR analysis. *OsACTIN* was used as an internal control. The data are mean ± SD (n = 3) and asterisks indicate the significant differences in different stages (** *p* < 0.01; Student’s *t*-test).

**Figure 4 plants-12-00409-f004:**
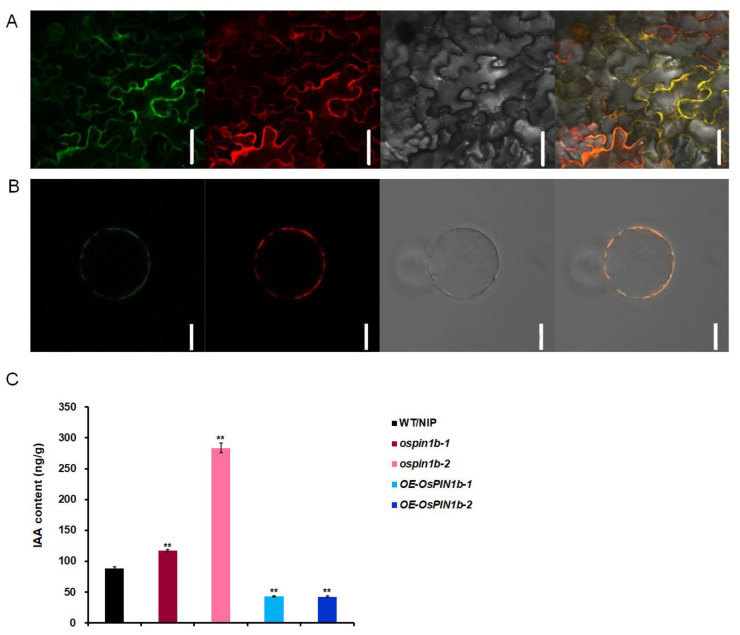
Subcellular localization of OsPIN1b and auxin contents in lamina joints. (**A**,**B**) *35S:OsPIN1b-sGFP* fusion construct and plasma membrane marker pm-rb*CD3-1008* were simultaneously co-expressed in *Nicotiana benthamiana* epidermal cells (upper) and rice protoplasts (lower). From left to right, represents green fluorescence of *35S:OsPIN1b-sGFP*, red fluorescence of membrane marker pm-rb*CD3-1008*, bright-field images, and yellow merged fluorescence, respectively. Scale bars = 10 μm. (**C**) Auxin contents in lamina joints of 7-day-old WT/NIP, *ospin1b-1, ospin1b-2, OE-OsPIN1b-1,* and *OE-OsPIN1b-2*. The three biological repeats were used in these experiments. The data are mean ± SD (n = 3) and asterisks indicate the significant differences in the above-mentioned lines (** *p* < 0.01; Student’s *t*-test).

**Figure 5 plants-12-00409-f005:**
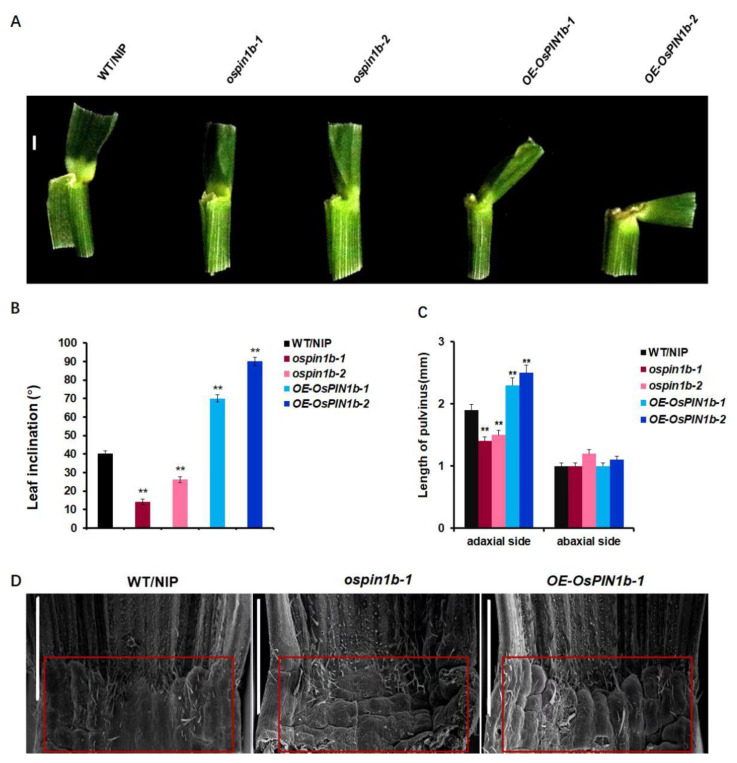
*OsPIN1b* regulates pulvinus. (**A**) Pulvinus of flag leaf of WT/NIP, *ospin1b-1*, *ospin1b-2, OE-OsPIN1b-1,* and *OE-OsPIN1b-2*. (**B**) The statistics analysis of flag leaf inclination in WT/NIP, *ospin1b-1*, *ospin1b-2*, *OE-OsPIN1b-1,* and *OE-OsPIN1b-2* lines. The data are mean ± SD (n = 3). The data are mean ± SD (n = 3) and asterisk indicates the significant differences (** *p* < 0.01; Student’s *t*-test). (**C**) The statistics analysis of the adaxial and abaxial side of pulvinus length in of WT/NIP, *ospin1b-1*, *ospin1b-2*, *OE-OsPIN1b-1,* and *OE-OsPIN1b-2*. Scale bar =1 mm. The data are mean ± SD (n = 3) and asterisk indicates the significant differences (** *p* < 0.01; Student’s *t*-test). (**D**) The adaxial surface of the pulvinus in WT/NIP, *ospin1b-1* and *OE-OsPIN1b-1* by SEM. The red box indicates cells on the adaxial surface of the pulvinus in WT/NIP, *ospin1b* and *OE-OsPIN1b*. Scale bar = 1 mm.

**Figure 6 plants-12-00409-f006:**
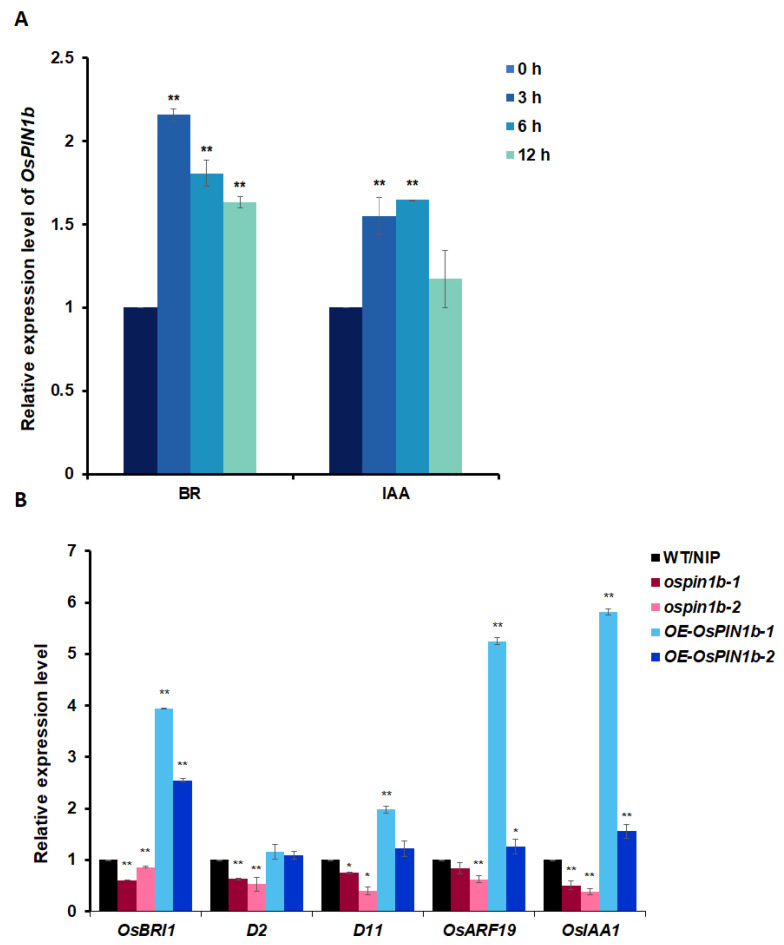
*OsPIN1b* regulates leaf inclination by activating BR and auxin signaling. (**A**) qRT-RCR analysis of the *OsPIN1b* expression under BR and IAA treatments with different time. The data are mean ± SD (n = 3), ** indicate significant difference at *p* < 0.01 (Student’s *t*-test). (**B**) The expression level of genes related to biosynthesis and signal transduction of BR and auxin in rice lamina joints of WT/NIP, *ospin1b-1*, *ospin1b-2, OE-OsPIN1b-1,* and *OE-OsPIN1b-2* by qRT-RCR. *OsACTIN* was used as an internal control. The data are mean ± SD (n = 3), * indicates significant difference at *p* < 0.05, and ** indicates statistical significance at *p* < 0.01 (Student’s *t*-test).

**Figure 7 plants-12-00409-f007:**
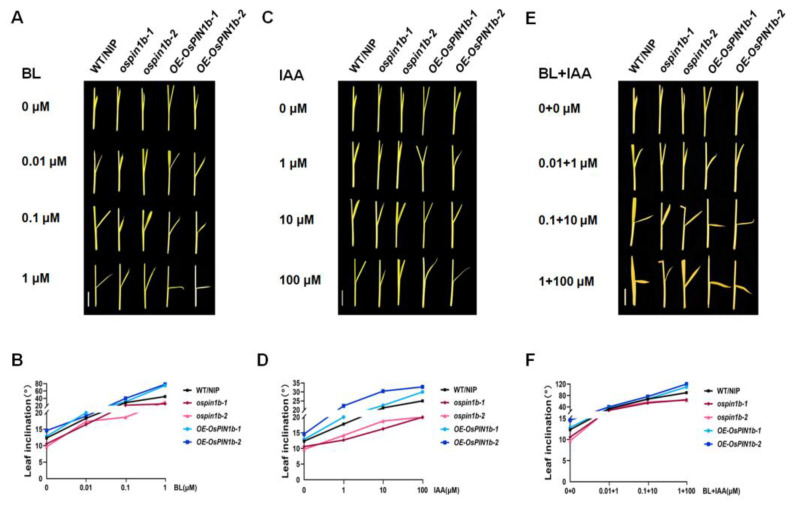
Variation in leaf inclination by BR and IAA treatments. (**A**) Leaf inclination response to 24-eBL treatment in WT/NIP, *ospin1b-1*, *ospin1b-2*, *OE-OsPIN1b-1,* and *OE-OsPIN1b-2* for 7-day-old seedling. (**B**) The statistical analysis of (**A**). (**C**) Leaf inclination response to IAA treatment. (**D**) The statistical analysis of (**C**). (**E**) Leaf inclination response to 24-eBL + IAA treatment. The concentrations of BL + IAA from top to bottom were 0 + 0, 0.01 + 1, 0.1 + 10, and 1 + 100 µM, respectively. (**F**) The statistical analysis of (**E**). The data in (**B**,**D**,**F**) are mean ± SD (n = 10 independent plants). Scale bars = 1 cm.

## Data Availability

All data supporting the findings of this study are available within the paper and within its Appendix A published online. The names of the repository/repositories and accession number (s) can be found in the article/Appendix A.

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
