# Peer review of "Auxin Transporter OsPIN1b, a Novel Regulator of Leaf Inclination in Rice (Oryza sativa L.)"

_plants, 2023, doi:10.3390/plants12020409_

Round 1

Reviewer 1 Report

This manuscript by Zhang et al. reveals the relationship between OsPIN1b and leaf inclination, and proves OsPIN1b responds to BR and auxin signaling. However, some points are needed to be addressed before further process.

 Q1:

Figure 2. Why the flag leaf inclination of the complementary ospin1b lines, C-ospin1b-1 and C-ospin1b-2, were significantly increased compared with the WT/NIP?

 Q2:

Figure 5. Please add a quantitative data to show that the adaxial cell number in the lamina joint of OE-OsPIN1b-1 is more than WT/NIP, because the number of cells (the red box in Figure 5D) on the adaxial surface of the pulvinus in OE-OsPIN1b-1 is not significantly more than WT/NIP.

 Q3:

1. Figure 6A. Under BR treatment with different time, was the Student's t-test right?

2. Figure 6B. Was there any more auxin related genes up-regulated in OE-OsPIN1b lines except OsIAA1 and OsARF19?

Q4:

Figure 7. Under non-BL or non-IAA treatment, the flag leaf inclination of WT/NIP, ospin1b mutants and OE-OsPIN1b were almost similar, which was different from the result of Figure 1C (the leaf inclinations in ospin1b mutants were reduced ~30% while in OsPIN1b overexpression lines was increased ~140% compared to WT/NIP). Please explain the reasons.

Q5:

Typo.

1. Line 116-118: “By statistical analysis, the leaf inclinations in ospin1b mutants was reduced ~30% while in OsPIN1b overexpression lines was increased ~140% compared to WT/NIP (Figure 1C,1D).” should be “By statistical analysis, the leaf inclinations in ospin1b mutants were reduced ~30% while in OsPIN1b overexpression lines were increased ~140% compared to WT/NIP (Figure 1C,1D).”

2. Line 246-247: “The results showed that OsPIN1b was induced by B and IAA” should be “The results showed that OsPIN1b was induced by BR and IAA”.

3. Line 421: “Each tissue (seeds germinated, 3 d; roots, 7 d; stems and leaves, 14 d; flowers), please also add the sampling period after “flowers”.

4. Line 429-431: “seedlings of wild-type, ospin1b mutant and OE-OsPIN1b) by using the TIANGEN RNAprep pure Plant Kit” should be “seedlings of wild-type, ospin1b mutant and OE-OsPIN1b by using the TIANGEN RNAprep pure Plant Kit”.

Reviewer 2 Report

Dear Authors, 

Review of the manuscript entited „Auxin Transporter OsPIN1b, a Novel Regulator of Leaf Inclination in Rice (Oryza sativa L.)” written by Yanjun Zhang et al..

In this manuscript the authors chracterized a novel regulator of rice  leaf inclination, auxin transporter OsPIN1b. They created two loss of function homozygous mutants, ospin1b-1 and ospin1b-2 with smaller leaf inclination as compared to the wild-type, Nipponbare (WT/NIP), while the overexpression lines, OE-OsPIN1b-1 and OE-OsPIN1b-2 have the opposite phenotype. Cell biological studies showed that in the adaxial region of OE-OsPIN1b-1 has significant bulge compared to WT/NIP and ospin1b-1, indicating that the increase of the adaxial cell division result in the enlarging of the leaf inclination in OE-OsPIN1b. The OsPIN1b was localized on plasma membrane. The free IAA contents in lamina joint of ospin1b mutants were significantly increased while they were decreased in OE-OsPIN1b lines, suggesting that OsPIN1b might action an auxin transporter like AtPIN1 to alter IAA content and leaf  inclination. Furthermore, the OsPIN1b expression was upregulated by exogenous epibrassinolide (24-eBL) and IAA, and ospin1b mutants are insensitive to BR or IAA treatment, indicating that they effecting leaf inclination is regulated by OsPIN1b. The ospin1b mutant had reduced plant height and leaf inclination which may contributes a new gene resource for molecular design breeding of rice architecture.

   I recommend this MS for acceptance after minor revision. The list of my recommendations are presented below.

2. Results

line 171:(I) qRT-PCR analysis of relative expression level of OsPIN1b in lamina joint at different stages of WT/NIP  time points.”

line 206: in Figure 4.: „(A, B) 35S:OsPIN1b-GFP fusion construct and plasma membrane marker pm-rbCD3-1008 were simultaneously co-expressed in tobacco Nicotiana benthamiana epidermal cells (upper)”. The tobacco word is equal with Nicotiana tabacum, but here you used Nicotiana benthamiana in your experiment.

line 247: „showed that OsPIN1b was induced by BR and IAA, indicating….”

line 296: Figure 6. What was used as an internal control? You shall have to indicate it here.

line 305: in the Figure 7. there are contradictory results presented compared with the Figure 1., where you presented the leaf inclination at normal stages. E.g. the leaf inclination in Figure 1. for the WT is 300, for the loss of funtion mutants are 20-250, for the overexpressing lines are 70-850 at ground stage (Figure 1D as  statistics). In the Figure 7., the leaf inclinations at 0 uM treatments are cc. 200 in Figure B, D, F. for every catergory. How can you explain this discrepancy? I would suggest to replace the rice plants photos at the 0 uM treatments in the Figure 7. for that one which represents the Figure 1. stage for the WT, mutants and overexpressing lines, respectively.

4. Mat&Methods:

line 385: the last Reference was in line 372 the Ref. 58.(„… planting lodging resistant wheat and rice semi dwarf varieties [58]”). So here can not come the Ref. 60., because the Ref. 59 is missing. You shall have to renumber from here the manuscript.

line 390: „10 μM of 24-eBL, epibrassinolide)

line 408: „…and ProOsPIN1b:GUS vectors were constructed as described previously [25].

line 414:” The 35S:OsPIN1b- 414 sGFP and plasma membrane marker pm-rbCD3-1008 were co-transfected into the tobacco Nicotiana bentamiana epidermal cells by Agrobacterium transformation…”

line 444: „Epibrassinolide (24-eBL; designated as BL; Solarbio Science and Technology, Beijing, China), was dissolved in DMSO” .

5. References:

General note for the References: you shall have to follow the Instructions for Authors for Plants for preparing the correct References like:

Journal Articles:
1. Author 1, A.B.; Author 2, C.D. Title of the article. Abbreviated Journal Name Year, Volume, page range.

·  Books and Book Chapters:
2. Author 1, A.; Author 2, B. Book Title, 3rd ed.; Publisher: Publisher Location, Country, Year; pp. 154–196.
3. Author 1, A.; Author 2, B. Title of the chapter. In Book Title, 2nd ed.; Editor 1, A., Editor 2, B., Eds.; Publisher: Publisher Location, Country, Year; Volume 3, pp. 154–196.

Sincerely yours,

Reviewer 1
